# NIR-Triggered Hyperthermal Effect of Polythiophene Nanoparticles Synthesized by Surfactant-Free Oxidative Polymerization Method on Colorectal Carcinoma Cells

**DOI:** 10.3390/cells9092122

**Published:** 2020-09-18

**Authors:** Deval Prasad Bhattarai, Beom Su Kim

**Affiliations:** 1Department of Chemistry, Amrit Campus, Tribhuvan University, Kathmandu 44613, Nepal; devalprasadbhattarai@gmail.com; 2Carbon Nano Convergence Technology Center for Next Generation Engineers (CNN), Chonbuk National University, 567 Baekje-daero, Deokjin-gu, Jeonju-si 54896, Korea

**Keywords:** polythiophene nanoparticles, chemical polymerization, photothermal, CT-26 cells, cancer treatment, apoptotic cell death

## Abstract

In this work, polythiophene nanoparticles (PTh–NPs) were synthesized by a surfactant-free oxidative chemical polymerization method at 60 °C, using ammonium persulphate as an oxidant. Various physicochemical properties were studied in terms of field emission scanning electron microscopy (FESEM), X-ray diffraction (XRD), Fourier transform infra-red (FT-IR) spectroscopy, and differential scanning calorimetry (DSC)/thermogravimetric analysis (TGA). Photothermal performance of the as-synthesized PTh–NPs was studied by irradiating near infra-red of 808 nm under different concentration of the substrate and power supply. The photothermal stability of PTh–NPs was also studied. Photothermal effects of the as-synthesized PTh–NPs on colorectal cancer cells (CT-26) were studied at 100 µg/mL concentration and 808 nm NIR irradiation of 2.0 W/cm^2^ power. Our in vitro results showed remarkable NIR laser-triggered photothermal apoptotic cell death by PTh–NPs. Based on the experimental findings, it is revealed that PTh–NPs can act as a heat mediator and can be an alternative material for photothermal therapy in cancer treatment.

## 1. Introduction

Cancer is a worldwide problem and poses huge challenges to human health. Among the different types of body cancer, colorectal cancer is the third most common cancer, and the most common cause of cancer-related morbidity [1]. Colorectal cancer is a heterogeneous tumor with different genetic alterations, which is associated with some kinds of gut microbiota and mutations targeting oncogenes, tumor suppressor genes, and gene related to DNA repair mechanism [2,3]. If not cured in time, the metastatic colorectal cancer may also lead to peritoneal carcinomatosis [4].

To date, various routes have been developed for the treatment of cancer, which include chemotherapy, immunotherapy, radiotherapy, thermal ablation, surgery, and regional combined therapy [5,6]. A combination of oxaliplatin with 5-fluorouracil and a gold-based anticancer drug, such as auranofin, are being used as adjuvant chemotherapy for colon cancer treatment [7,8]. However, these modalities are less effective at addressing the case of metastasis. All in all, development of less aggressive and more effective therapies, compared to those of conventional ones, is necessary in the treatment of colorectal cancer. Since the last decade, cancer treatment by carcinoma cell ablation methods is attracting more research attention [9,10]. Radiofrequency ablation (RFA), microwave ablation (MWA), and laser ablation (LA) methods have been shown to be more effective in the cancer cell treatment by thermal ablation modality [11]. The ablation method offers benefit over conventional methods of cancer treatment, as it shows good local tumor control with minimum complication rates. This method outsmarts other common methods by avoiding the nuisances of multiple surgery in the case of metastasis. Furthermore, the thermal ablation method is attracting more interest, as it can be performed on an outpatient basis.

In this context, photothermal therapy could be an outstanding technique for cancer cell ablation, as it is regarded as a highly selective, minimally invasive, and remotely controllable thermal ablation technique [9]. For this purpose, near infra-red radiation of specified power is irradiated over the cancer cell adjunct with photothermal materials for a specified time of exposure. The first near infra-red (NIR) window falls in the range of 700–900 nm, which possesses effective tissue penetration ability, and can lead to selective cancer cell ablation in the presence of accumulated photothermal agents.

Previously, inorganic-based photothermal agents, such as CuS and Au NPs, were used for cancer cell ablation purpose. Study shows that metallic oxides or compounds are more likely to be toxic [12,13]. Later, researches on organic compound-based photothermal agents were reported. Such organic compound based photothermal agents are more tunable, and can be more functionalized, compared to inorganic-based photothermal agents. Currently, organic conjugated polymers, such as polypyrrole, polyaniline, and polythiophene and their derivatives, are gaining more preference to inorganic-based photothermal agents, due to their ease of synthesis, tunable surface functionality, and biocompatibility aspects [14,15]. In addition to chemical nature, geometry of nanomaterials also plays a crucial role in photothermal efficacy. Photothermal agents with higher surface area avail more irradiation space and exhibit higher thermal ablation efficacy upon NIR irradiation of specific power supply, and irradiation time over a specific concentration of nanomaterial concentration [9].

Among the various pre-requisites of a material to be used for biomedical applications, the aspect of biocompatibility is most important one. Some conjugated conductive polymers such as polyaniline, polypyrrole, polythiophene, etc. are reported to be biocompatible [16]. Among these polythiophene is least studied conductive polymer and its in vitro biocompatibility towards bone cells and nerve cells has been revealed [14]. In this context, this work has been focused to study the Polythiophene nanoparticle as a potential candidate for a photothermal agent.

Different routes of synthesis, including electrochemical, chemical, and enzymatic polymerization methods, have been reported for the synthesis of polythiophene [17,18,19,20]. PTh–NPs have been synthesized by cationic surfactant-assisted dilute polymerization of thiophene, using ferric chloride as an oxidant [21,22]. Jung et al. synthesized PTh–NPs via Fe^+++^/H_2_O_2_ catalyzed oxidative polymerization in aqueous medium [23]. Anionic surfactant-assisted and Cu(II) catalyzed oxidative polymerization of PTh–NPs in aqueous media showed the formation of spherical PTh–NPs [24]. Recently, coating of polythiophene onto the surface of gold by a biphasic liquid–liquid interfacial polymerization route has been reported [25,26]. In some cases, surfactant-assisted polymerization may add some doped ions onto polymer which, in some cases, may influence on material conductivity or in some other cases may induce some sorts of cytotoxicity. Under these considerations, we designed the synthesis of PTh–NPs without the use of surfactant. This process also enjoys the benefits of lessening the route of purification.

In this work, PTh–NPs were synthesized by oxidative chemical polymerization of thiophene in the presence of ammonium persulphate as an oxidant under surfactant-free condition. The photothermal effects of the as-synthesized PTh–NPs on colorectal cancer cells (CT-26) were studied at 100 µg/mL concentration and of NIR irradiation of 2 W/cm^2^ power. The photothermal stability of PTh–NPs was also studied.

## 2. Materials and Methods

### 2.1. Materials

Thiophene (purity ≥ 99%), ammonium persulphate (purity ˃ 98%), and methanol (purity 99.5%), ethanol (purity 95%) were purchased from Sigma-Aldrich (St. Louis, MO, USA). Acetonitrile (purity ˃ 99.5%), was purchased from Samchun (Seoul, Korea). All the materials and reagents were of analytical grade, and were used as received, without any further purification.

### 2.2. Preparation of Polythiophene Nanoparticles

Polythiophene nanoparticles were synthesized according to our previously reported methods [27]. Briefly, 20 mL of semi molar ammonium persulphate solution was dropped into a vessel including 30 mL of semi molar (M/2) thiophene solution in decimolar (M/10) acetonitrile solution in aqueous media with constant stirring. The mixture of solution was heated at 60 °C with constant magnetic stirring at 300 relative centrifugal forces (RCF) for 24 h. The solution was then taken out and allowed to settle for 12 h under ambient condition. Then it was washed with methanol, shaking in shaker each sample for half an hour followed by centrifugation at 5000 RFC for 10 min (Beckman Optima TM LE-80K, Beckman Inc., Brea, CA, USA). Then the supernatant solution was discarded keeping the solid residue in the tube. The process was repeated some more times. The thus obtained PTh substrate was first dried at room temperature (RT), and dried in vacuum drier for 6 h at 60 °C. This PTh was used for characterization, and for various tests.

### 2.3. Physico-Chemical Characterization

The surface morphology and topography of the as-synthesized polythiophene was studied using Field-emission scanning electron microscopy (Zeiss SUPRA 40 VP FESEM, Cal Zeiss SMT Inc., Peabody, MA, USA) at an accelerating voltage of 5.0 kV. Particle dimensions were measured using Image J software (NIH, Bethesda, MD, USA) from different images of the same sample to obtain average dimension. The particle size distribution was determined by dynamic light scattering (DLS) analysis using Zetasizer ZS90 (Malvern Instruments Ltd., Malvern, UK). Biological Transmission electron microscopy (Bio-TEM, Hitachi H-7650, Hitachi, Tokyo, Japan) was used to inspect the morphology of the as-synthesized PTh–NPs under an accelerating voltage of 100 kV. For the TEM analysis, PTh–NPs were dispersed into ethanol solution, and one drop was casted over C-flat™ carbon grids (Electron Microscopy Science, Hatfield, PA, USA). X-ray diffraction (XRD) patterns were obtained using Rigaku X-ray diffractometer (Japan) with Cu–Kα (λ = 1.54Å) radiation over Bragg’s angle (20) of (5–90)°. Polymerization of thiophene into polythiophene was studied using Spectrum GX Fourier transform infra-red (FT-IR) spectroscopy (PerkinElmer, Inc., Shelton, CT, USA). FT-IR spectra were measured in the range (4000–400) cm^−1^. Thermal stability of the as-synthesized PTh–NPs was studied using thermogravimetric analysis (TGA) and differential scanning calorimetry (DSC).

### 2.4. Photothermal Properties of Polythiophene Nanocomposites

Photothermal behavior of PTh–NPs was studied by recording the temperature increase of aqueous medium containing dispersed PTh–NPs (100 µg/mL) under the illumination of NIR laser (808 nm). The test was carried out by putting 1 mL PTh–NPs dispersed aqueous solution in 48-well plate, and it was subjected to NIR laser intensities of 2.0 W/Cm^2^ for 5 min, to study the power-dependent photothermal effects on the substrate. Real-time temperature changes exhibited by the sample were recorded by FLIR camera (FLIR Systems Pty Ltd, Sydney, Australia). Temperature change of water under NIR irradiation was taken as control. Photothermal stability was studied by recording the temperature change of the sample dispersed in aqueous solution over 10 cycles under the illumination of NIR radiation of 2.0 W/Cm^2^ for 5 min NIR on, followed by 5 min NIR off.

### 2.5. Cell Culture

Murine colorectal cancer cells (CT–26) were purchased from the American Type Collection (ATCC) and used for cancer cell model in this study. The cells were cultured in RPMI1640 (Thermo Fisher Scientific, Waltham, MA, USA) medium supplied with 10% fetal bovine serum (Thermo Fisher Scientific) and 1% antibiotics (Thermo Fisher Scientific) at 37 °C under 5% CO_2_. The cell culture medium was refreshed every 2 days during the experiment.

### 2.6. Cytotoxicity

Cell viability was determined by CCK-8 assay. Briefly, the CT–26 cells (5 × 10^3^) were seeded in 96-well culture plate and cultured overnight. Then NPs were added to each well at various final concentration of (25, 50, 100, 200, and 500) µg/mL. After 24 h of cultivation, cells were washed with PBS. 20 μL of CCK-8 kit (Sigma–Aldrich) solution and 200 μL of RPMI 1640 cell culture medium (Thermo Fisher Scientific) were added to each well and incubated for 1 h at 37 °C. Plates were then analyzed with a microplate reader (Tecan Group Ltd., Salzburg, Austria) at 450 nm.

### 2.7. Cellular Uptake Assay

To determine the cellular uptake of NPs, various concentrations of NPs were added to each well. After 4 h of cultivation, cells were washed with PBS 3 times, and fixed with 4% paraformaldehyde. The cells were then observed by microscopy (Eclipse Ts2-FL, Nikon, Tokyo, Japan).

### 2.8. Hyperthermia Effects of PTh–NPs

The CT–26 cells were seeded into 48-well plate and incubated for 24 h. Then the PTh–NPs were added into each well at 100 μg/mL of final concentration of NPs for 4 h. Next, CT–26 cells were exposed to an 808 nm laser (2.0 W/cm^2^) for 5 min and incubated again for 24 h. The viability was measured by CCK-8 assay. To confirm the viability, additional Live/dead staining assay was performed [9]. Briefly, after 24 h of cultivation, cells were rinsed with PBS and the Live/Dead^®^ Viability/Cytotoxicity kit (Molecular Probe, Eugen, OR, USA) reagent solution was added. After incubation for 30 min in CO_2_ incubator at 37 °C, the cells were observed [28] using an inverted fluorescence microscope (Eclipse Ts2-FL).

### 2.9. In Vitro Cell Apoptosis Analysis

To determine if NIR triggered CT–26 apoptosis, CT–26 cells were seeded in 48-well plate at 1 × 10^4^ per well and incubated with PTh–NPs. Cells were then irradiated with 808 nm laser (2.0 W/cm^2^, 5 min). After incubation for another 24 h, the cells were collected by trypsinization, and stained by Annexin V-FITC and PI, according to the manufacturer’s protocol. To detect the apoptotic cells, flow cytometry (CytoFlex, Beckman, IL, Milan, Italy) were used at a 488 nm. Laser was used to excite both dyes; the FL1 channel was used to collect the emission of Annexin V-FITC, and the FL2 channel was used to collect the emission of PI.

### 2.10. Statistical Analysis

All experiments were performed in triplicates. Values are expressed as mean ± standard deviation. Statistical analysis was performed by one-way analysis of variance followed by Tukey-test using GraphPad Prism software (San Diego, CA, USA). *P* < 0.05 was considered statistically significant.

## 3. Results and Discussion

### 3.1. Physicochemical Characterization

The general structure and surface morphology of the as-synthesized PTh–NPs were studied with the help of FESEM (Figure 1a). The imagery clearly reveals the spherical aggregation of PTh–NPs. The elemental analysis of PTh–NPs was carried out using energy-dispersive X-ray spectroscopy (EDX) (Figure 1b). The presence of carbon and sulphur in EDX mapping ensures that these elements are the constituents of PTh–NPs. The crystallinity of the as-synthesized PTh–NPs was studied in terms of X-ray diffraction imagery, which shows a low range of crystallinity. The particles size distribution and size of PTh-NPs were determined by DLS (Figure 1c) and Bio-TEM (Figure 1d) analyses. DLS results showed that the narrow size distribution of particles with average size of ~69.8 nm. In addition, the observed TEM image was consistent with particle size by DLS.

### 3.2. Thermogravimetric (TGA) and Differential Scanning Calorimetry (DSC) Study

Thermal behavior, like weight loss as a function of heat, residue content, and decomposition pattern of PTh–NPs, were studied in terms of thermogravimetric analysis (TGA) and differential scanning calorimetry (DSC). Around 96% mass retention was observed at about 200 °C. Heating the substance above this temperature exhibited a significant amount of mass loss. At about 800 °C, the residue content was around 36%. These data show the significant thermal stability of the as-synthesized PTh–NPs for its biomaterial application, where the material is expected to withstand autoclaving temperature, though these are targeted to work at body temperature. The small quantity of mass loss of PTh–NPs at up to around 200 °C could be associated with the loss of volatile impurities and inherent moisture (Figure 2a). The differential scanning curve (DSC) shows a glass transition temperature (T_g_) of 51 °C. The endothermic peak in the DSC thermogram is observed at 257 °C. The T_g_ value above room temperature signifies the rigid structural properties of the as-synthesized PTh–NPs (Figure 2b).

### 3.3. Photothermal Analysis and Photothermal Stability of PTh–NPs

Photothermal therapy (PTT) is a physicochemical process, in which incident radiation falling onto some kinds of material results in an increase in temperature, a condition of hyperthermia. In the present study, near infra-red (NIR) of 808 nm wavelength was passed onto PTh-NPs dispersed in deionized water for specified period of time under different power supply, and the resulting temperature was noted. As a negative control, deionized water was used for photothermal study, where the resulting temperature was noted to be in the range 30–31 °C. As expected, the temperature was found to increase with the increase of irradiation power (Table 1) and (Figure 3a). Furthermore, the temperature was found to increase with increase in concentration of PTh-NPs dispersion under the same power irradiation (Table 1). For example, for the irradiation of 2.0 W/cm^2^ NIR 808 nm, the maximum temperature recorded for PTh-NP at 100 µg/mL and 200 µg/mL are 45.1 °C and 49.4 °C, respectively. The result shows that the increase in temperature is directly related to the concentration of the substrate. The temperature around or above 45 °C is sufficient to ablate the cancerous cells by hyperthermia [9,29], which can be obtained by tuning the substrate concentration and irradiation power.

Photothermal stability is a very important aspect of material used for hyperthermia [30]. For this purpose, the photothermal stability of PTh–NPs was studied. PTh–NPs of 100 µg/mL concentration was taken, and was subjected to NIR (808 nm) irradiation for 5 min at power of 2.0 W/cm^2^, followed by cooling for 5 min at RT. The cycle was repeated 10 times. The highest temperature recorded for these 10 cycles was almost constant, which confirms the substrate exhibits excellent photothermal stability (Figure 3b).

### 3.4. In Vitro Cell Ablation Test

Biocompatibility is the most important aspect while considering the nanoparticles for biomedical application [31]. Therefore, we first determined the effects of PTh–NPs on cell viability. CT-26 cells were incubated with PTh–NPs for 24 h, and cell viability was evaluated. The results of CCK-8 assay showed that PTh–NPs are non-cytotoxic to CT-26 cells at 25–250 µg/mL concentration. But high concentration (500 µg/mL) of PTh–NPs exhibited little cytotoxicity (Figure 4a). Crystal violet staining result (Figure 4b) was also consistent with the CCK-8 results. These results indicate that PTh–NPs in the range 25 to 250 µg/mL concentration were adaptable for further biomedical application.

After treatment with PTh–NPs for 3 h, cellular uptake of PTh–NPs was characterized by dark-field microscopy (Figure 5a). The microscopy results showed that after the treatment with PTh–NPs, the PTh–NPs were scattered in the cytoplasm. Furthermore, the cellular uptake results indicated that the uptake of PTh–NPs also depended on the concentration of nanoparticles [32].

To evaluate the NIR laser-triggered photothermal cancer cell ablation by PTh–NPs, the PTh–NPs treated CT-26 cells were exposed to irradiation for 5 min using an 808 nm laser at a power density of 2.0 W/cm^2^ as an optimal condition. By treatment with NIR, around 45.1 °C temperature was observed in the cell culture well that was incubated with PTh–NPs at 100 µg/mL (Figure 5b). Live/dead staining results showed that the number of dead cells (stained red) increased significantly for CT-26 cells that had been incubated with PTh–NPs. In contrast, from groups of CT-26 cells only, NIR irradiation only, and incubation with PTh–NPs only, very few dead cells were observed (Figure 5c). Furthermore, CCK-8 assay showed that the cell viability of CT-26 cells significantly decreased at a PTh–NPs treated cell by NIR irradiation photothermal ablation (Figure 6a). Next, we used flow cytometry analysis to evaluate the apoptotic cell death effect of PTh–NPs treated with laser irradiation by Annexin V-FITC/PI assay (Figure 6b,c). Phospholipids (PS) translocation of the outer layer of plasma membrane is a characteristic of the early phase of apoptosis. Annexin V has a high affinity for PS and therefore it is generally used to detect apoptotic cell death. However, Annexin V can also stain necrotic cells. Therefore, cells were double stained with propidium iodide (PI) to distinguish between the viable cells (Annexin V^−^/PI^−^) early apoptotic cells (Annexin V^+^/PI^−^), late apoptotic cells (Annexin V^+^/PI^+^), necrotic cells (Annexin V^−^/PI^+^) [33]. The FACS analysis result is shown in Figure 6b, the four-quadrant plots in each panel show the viable cells (lower left), early apoptotic cell (lower right), late apoptotic cells (upper right) and necrotic cells (upper left). The FACS analysis result showed more than 90% observed viable cells in the control, and only PTh–NPs treated groups. In only NIR treated group, a little amount late apoptotic cells (5.9%) were detected. However, after 808 nm laser irradiation, the PTh–NPs treated cell groups showed the early apoptotic (23.62%), late apoptotic cells (23.76%) and necrotic cell death (13.8%). The percentage of apoptotic/necrotic cells dramatically increased in PTh-NPs/NIR 808 nm group to compare other groups.

Hyperthermia can induce endoplasmic reticulum (ER) stress and result in apoptosis [34] in numerous cancers. In addition, hyperthermia induces reactive oxygen species (ROS) [35] and the functional disorders of the mitochondria [36]. Although the apoptotic molecular mechanism was not revealed in this study, our PTh–NP/NIR trigged apoptotic cell death is maybe related to ER stress and/or disorder of mitochondria function mechanism. Therefore, these results indicate that the PTh–NPs prepared by a surfactant-free oxidative polymerization method could be used as a heat mediator for the hyperthermia treatment of cancer cells.

## 4. Conclusions

In summary, we successfully prepared PTh–NPs by a surfactant-free oxidative polymerization method. The PTh–NPs exhibited photothermal effect under the NIR 808 irradiation. Furthermore, the as-synthesized PTh–NPs demonstrated good photothermal stability, which could be linked to the material stability for its destined application. In vitro cell viability cell showed that the as-synthesized PTh–NPs were non-cytotoxic to CT-26 cells at (25–250) µg/mL concentration. NIR laser-triggered photothermal treatment of the cancer cells incubated with PTh–NPs showed increased number of dead cells, compared to those without PTh–NPs. The flow cytometry result showed that the PTh–NPs -treated cell groups showed a remarkable cell death effect. These results suggest a route for the use of PTh–NPs as a heat mediator for the hyperthermia treatment of cancer cells. Although in vivo experiments were not conducted in this study, it is generally known that nanoparticles pass through the wall of loosed blood vessel of cancer tissues and gets accumulated. Therefore, based on these findings, the PTh–NPs can be a good candidate for use as a composite material to induce and enhance further important and desired properties. Furthermore, bioactive molecules can be incorporated into such kinds of nanoparticles to enhance important properties, to induce novel properties that target biomedical applications. This finding suggests a promising research area of the use of conjugated polymeric nanomaterials in the field of biomedical applications. Further optimization, development, and extension of the present work would be considered in the continuation of this work in the near future.

## Figures and Tables

**Figure 1 cells-09-02122-f001:**
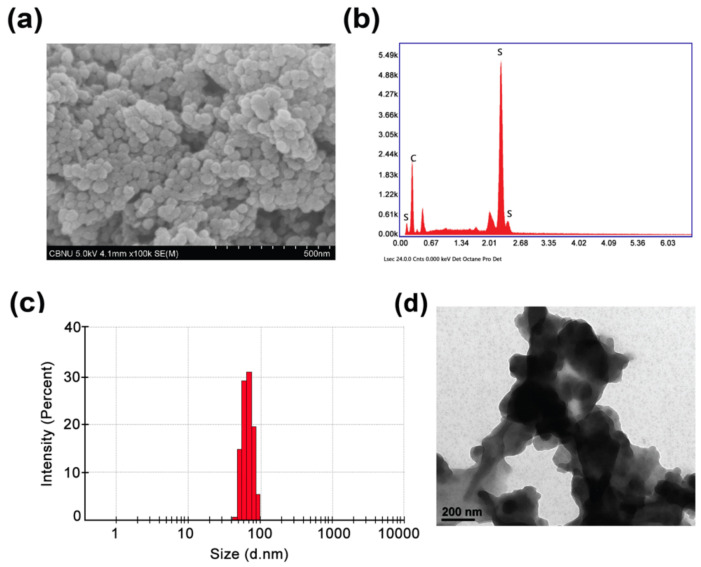
Physicochemical characterization of polythiophene particles. (**a**) FESEM image, and (**b**) EDX mapping of PTh–NPs, which shows the presence of C and S elements, which are the constituent elements of polythiophene. Particles size dirstribution diagram by (**c**) DLS anaysis and (**d**) by Biological Transmission Electron Microscopy.

**Figure 2 cells-09-02122-f002:**
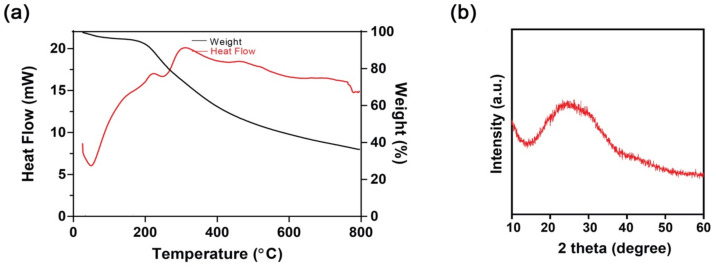
Characterization of PTh–NPs of (**a**) Thermogravimetric curve (wt. % vs. temperature curve) and differential scanning curve (heat flow vs. temperature curve, and (**b**) X-ray diffraction.

**Figure 3 cells-09-02122-f003:**
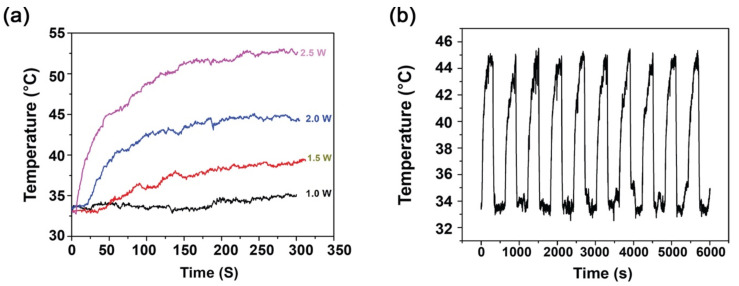
Photothermal study of PTh–NPs. (**a**) Photothermal performance of polythiophene nanocomposites, and (**b**) photothermal stability curve by real time temperature record for PTh–NPs, 100 µg/mL, under irradiation of 2.0 W/cm^2^.

**Figure 4 cells-09-02122-f004:**
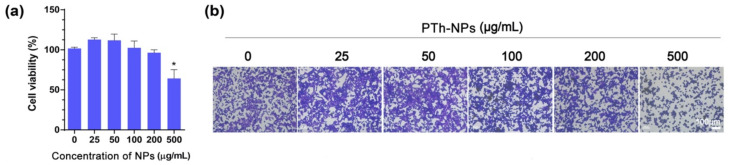
Cytotoxicity study of PTh–NPs. (**a**) Cell viability and (**b**) Crystal violet stained microscope images of CT-26 cell by treatment with PTh–NPs at concentration of (25, 50, 100, 200, and 500) μg/mL. Each value is expressed as the mean ± standard deviation. * *P* < 0.05, when compared with the control.

**Figure 5 cells-09-02122-f005:**
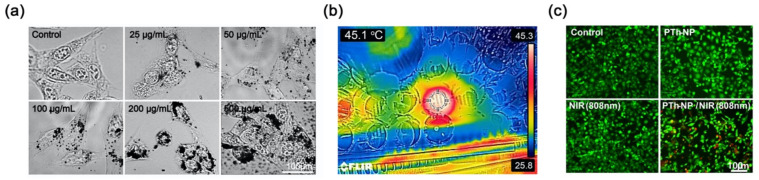
In vitro cellular uptake and photothermal effects of PTh–NPs. (**a**) Cellular uptake image of PTh–NPs in CT-26 cells using microscopy after 3 h of incubation. (**b**) Infrared photothermal image of PTh–NPs (100 μg/mL) dispersed in CT-26 cells culture media under 808 nm laser irradiation after 5 min at 2.0 W/cm^2^. (**c**) Live/dead fluorescence images of PTh–NPs treated CT-26 cells with/without 808 nm laser irradiation (2.0 W/cm^2^, 5 min).

**Figure 6 cells-09-02122-f006:**
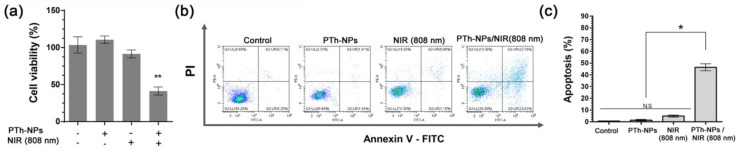
Photothermal effects of PTh–NPs on apoptotic cell death. (**a**) Cell viability of PTh–NPs (100 μg/mL) treated CT-26 cells with/without 808 nm laser irradiation (2.0 W/cm^2^, 5 min) at 24 h of cultivation. (**b**) Flow cytometric analysis to determine cell death of CT-26 cells after PTh–NPs with/without NIR irradiation. (**c**) Percentage of apoptotic CT-26 cells. Each value is expressed as the mean ± standard deviation. * *P* < 0.05 and ** *P* < 0.01, when compared to the control.

**Table 1 cells-09-02122-t001:** Photothermal ability of Polythiophene nanoparticles under the 808 nm NIR irradiation at different concentration.

Concentration of PTh–NPs (µg/mL)	Maximum Temperature (°C) Recorded under Different Power Supply of NIR 808 nm for Different Concentration of Substrate			
	1.0 W/cm^2^	1.5 W/cm^2^	2.0 W/cm^2^	2.5 W/cm^2^
100	35.0	39.2	45.1	53.5
200	36.0	42.5	49.4	57.7

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
