# Peer review of "NIR-Triggered Hyperthermal Effect of Polythiophene Nanoparticles Synthesized by Surfactant-Free Oxidative Polymerization Method on Colorectal Carcinoma Cells"

_cells, 2020, doi:10.3390/cells9092122_

Round 1

Reviewer 1 Report

The authors described the synthesis and application of polythiophene nanoparticles as a novel photothermal agent for photothermal therapy. The main idea of work are interesting, however I have some comments for authors:

  1. In my opinion, Figure 1 does not present nanoparticles, but some undefined,, composite'' without any specific size and shapes. Therefore, maybe authors should consider calling the product as nanocomposite.
  2. The authors presented the high activity of polythiophene NPs but do not describe any mechanism. 
  3. Please explain Figure 4. How cell viability could be higher than 100%? Maybe the applied method for the cell viability test is not sufficient for this purpose.
  4. In my opinion, the determination of the laser power as ,,2W/cm2'' is ambiguous. I know that this is a typical parameter, but it could be useful for readers if authors could determine the effective size of laser spot beam and power of this beam.

Author Response

Response to Reviewer 1 Comments

The authors described the synthesis and application of polythiophene nanoparticles as a novel photothermal agent for photothermal therapy. The main idea of work are interesting, however I have some comments for authors:

1. Reviewer’s comment: In my opinion, Figure 1 does not present nanoparticles, but some undefined,, composite'' without any specific size and shapes. Therefore, maybe authors should consider calling the product as nanocomposite.

Author’s Response: Thank you Reviewer for pinpointing. In this revised work, the particle size of as-synthesized polythiophene was measured by DLS method and the average size of as-synthesized polythiophene nanocomposite was recorded about ~69.8 nm.

2. Reviewer’s comment: The authors presented the high activity of polythiophene NPs but do not describe any mechanism. 
Author’s Response:  In general, hyperthermia-induced cancer cell death is caused by high temperature. This study presents the possibility of their effects by NIR and PT-NCs. Actually photons of NIR fall into polythiophene nanocomposites. Excitation of molecules takes place followed by non-radiative transition which is the cause of increase in temperature upon NIR irradiation. This phenomenon is more pronounced in conductive polymers like polyaniline, polythiophene, and polypyrrole due to the availability of delocalized electrons. This is the theoretical concept of hyperthermia, regarding to photochemistry. Being more theoretical, we have just presented a brief in the text.

3. Reviewer’s comment: Please explain Figure 4. How cell viability could be higher than 100%? Maybe the applied method for the cell viability test is not sufficient for this purpose.

Author’s Response: Cell viability was calculated compared with the control group. Cell viability of control group was taken as 100%. Survival rate more than 100% means cell proliferation was induced by the PTh-NCs.

4. Reviewer’s comment: In my opinion, the determination of the laser power as, 2W/cm2'' is ambiguous. I know that this is a typical parameter, but it could be useful for readers if authors could determine the effective size of laser spot beam and power of this beam.

Author’s Response: Thank you reviewer for your comment and suggestion. NIR radiation was irradiated into a well of 48-well plate keeping 1 mL water and different power of NIR 808 nm was passed into the solution just 5 cm above the surface of liquid.

Reviewer 2 Report

In his manuscript, Beom Su Kim described the preparation and characterization of polythiophene NPs used for NIR-triggered hyperthermal effect on colorectal carcinoma cells.

Despites the interest of the obtained results, the author needs to address the following questions before considering the publication of this manuscript in Cells. In other words, I do recommend the publication of B.S. Kim manuscript in Cells after major revisions.

Please find below the questions that need to be addressed:

  1. Abstract, page 1: The two last sentences of the Abstract are quite similar, the author can fuse them.
  2. Introduction, page 1 line 36: Does the author consider all the cancer types or only the colorectal one?
  3. Introduction, page 1 lines 38-39 “A combination of … treatment.”: The author has to precise for which kind of cancer this treatment is used and add at least a reference.
  4. Introduction, page 2 lines 59-61: The author has to give corresponding references.
  5. Introduction, page 2 lines 67-73: This paragraph needs to be reworded, and probably re-organized. Indeed, while the two first sentences concern the influence of geometric properties of photothermal compound on their activities, the two last sentences concern the biocompatibility. I do not see the link between them.
  6. Introduction, page 2 line 15: The reference 15 concerns only enzymatic polymerization, please add references concerning the other types of polymerization.
  7. Materials and Methods, page 3 line 99: I guess that the word “having” has to be hanged to “containing”.
  8. Materials and Methods, page 3 line 122: The words “the elevated temperature” have to be changed to “the temperature increase”.
  9. Materials and Methods, page 4 line 139: What is CCK-8 assay? The author has to describe briefly this assay and give the corresponding reference(s).
  10. Materials and Methods, page 4 line 144: The author has to precise the kind of microscopy and the type of microscope used for cellular uptake assay.
  11. Materials and Methods, page 4 lines 148-149: The sentence “The viability … respectively.” Has to be reworded. What is “Live/dead staining assay”? As for the CCK-8 assay, the author has to briefly describe it and gives the corresponding reference(s).
  12. Results and Discussion, page 4 lines 157-159: This sentence has to be erased.
  13. Results and Discussion, page 4 lines 167-168 “The average size … (Fig. 1c).”: To my opinion, the TEM images showed in Figure 1c are not enough clear to measure NPs diameters. Why doesn’t the author use dynamic light scattering (DLS) to measure NPs diameter?
  14. Results and Discussion, page 4 legend of Figure 1: What is Bio-TEM?
  15. Results and Discussion, page 5 line 191: I guess that the word “result” has to be changed to “results”.
  16. Results and Discussion, page 5 lines 198-199: The author has to link this sentence with their results. Do they obtain the expected system? If yes, which is/are the selected system(s).
  17. Results and Discussion, page 6 lines 214-215: To my opinion, the sentence “Biocompatibility … application [24].” needs to be reworded.
  18. Results and Discussion, page 6 line 217: To my opinion, “exhibited non-cytotoxicity” has to be changed to “exhibited no cytotoxicity” or “are non-cytotoxic”.
  19. Results and Discussion, page 6 lines 218-219: Where is described this crystal violet staining experiment? Why does the author conclude that the crystal violet staining result was consistent with the CCK-8 results? The author has to develop and to explain his results and why they are consistent.
  20. Results and Discussion, page 7 lines 241-243: The sentence “Furthermore … ablation (Fig 6a).” needs to be slightly modified.
  21. Conclusions, page 7: I agree that the encapsulation of bioactive molecules into NPs can be important and interesting. However, why does the author want to encapsulate a bioactive molecule and to which kind of bioactive molecule the author think?

Author Response

Reviewer #2

In his manuscript, Beom Su Kim described the preparation and characterization of polythiophene NPs used for NIR-triggered hyperthermal effect on colorectal carcinoma cells.

Despites the interest of the obtained results, the author needs to address the following questions before considering the publication of this manuscript in Cells. In other words, I do recommend the publication of B.S. Kim manuscript in Cells after major revisions.

Please find below the questions that need to be addressed:

  1. Reviewer’s comment: Abstract, page 1: The two last sentences of the Abstract are quite similar, the author can fuse them.
    Author’s Response: Thank you reviewer for your suggestion. In this revised manuscript, we have fused the sentences carrying common theme.

  2. Reviewer’s comment: Introduction, page 1 line 36: Does the author consider all the cancer types or only the colorectal one?
    Author’s Response: Thank you reviewer for the comment. Basically, we consider colorectal cancer. However, its metastatic proliferation may lead to other associated cancer. In this article, CT26 has been used as a cancer cell model. We need to experiment for other types of cancer cells which will be considered in successive works.

  1. Reviewer’s comment: Introduction, page 1 lines 38-39 “A combination of … treatment.”: The author has to precise for which kind of cancer this treatment is used and add at least a reference.
    Author’s Response: Thank you Reviewer for pinpointing. We have precisely mentioned about colorectal cancer and have added two relevant citations.

  2. Reviewer’s comment: Introduction, page 2 lines 59-61: The author has to give corresponding references.
    Author’s Response: Thank you reviewer for your suggestions. We have given the references in the relevant place.

  3. Reviewer’s comment: Introduction, page 2 lines 67-73: This paragraph needs to be reworded, and probably re-organized. Indeed, while the two first sentences concern the influence of geometric properties of photothermal compound on their activities, the two last sentences concern the biocompatibility. I do not see the link between them.

    Author’s Response: Thank you Reviewer for your suggestions. We have reworded, re-organized into separate paragraph with sequential link.

  1. Reviewer’s comment: Introduction, page 2 line 15: The reference 15 concerns only enzymatic polymerization, please add references concerning the other types of polymerization.
    Author’s Response: Thank you reviewer for your suggestions. We have added some more relevant references there in.

  2. Reviewer’s comment: Materials and Methods, page 3 line 99: I guess that the word “having” has to be hanged to “containing”.
    Author’s Response: Thank you reviewer for your suggestion. We have changed it.

  3. Reviewer’s comment: Materials and Methods, page 3 line 122: The words “the elevated temperature” have to be changed to “the temperature increase”.
    Author’s Response: Thank you Reviewer for your comment. We have changed the phrase according to your suggestion.

  4. Reviewer’s comment: Materials and Methods, page 4 line 139: What is CCK-8 assay? The author has to describe briefly this assay and give the corresponding reference(s).
    Author’s Response: Thank you Reviewer for your suggestions. We have further briefly described the CCK-8 assay method in this revised manuscript.

  5. Reviewer’s comment: Materials and Methods, page 4 line 144: The author has to precise the kind of microscopy and the type of microscope used for cellular uptake assay.
    Author’s Response: Thank you Reviewer for your suggestions. We have mentioned the kind of microscopy used.

  6. Reviewer’s comment: Materials and Methods, page 4 lines 148-149: The sentence “The viability … respectively.” Has to be reworded. What is “Live/dead staining assay”? As for the CCK-8 assay, the author has to briefly describe it and gives the corresponding reference(s).
    Author’s Response: Thank you reviewer for your comment. We have further briefly described the Live/Dead staining assay method.

  7. Reviewer’s comment: Results and Discussion, page 4 lines 157-159: This sentence has to be erased.
    Author’s Response: Thank you reviewer for your comment. The sentence had been deleted.

  8. Reviewer’s comment: Results and Discussion, page 4 lines 167-168 “The average size … (Fig. 1c).”: To my opinion, the TEM images showed in Figure 1c are not enough clear to measure NPs diameters. Why doesn’t the author use dynamic light scattering (DLS) to measure NPs diameter?
    Author’s Response: Thank you reviewer for your comment. According to your suggestion, we used DLS to measure the NPs diameter. We have mentioned in the revised manuscript.

  1. Reviewer’s comment: Results and Discussion, page 4 legend of Figure 1: What is Bio-TEM?
    Author’s Response: Thank you reviewer for your comment. Bio-TEM is biological transmission electron microscopy. This type of TEM is used to observe the biological specimen.

  2. Reviewer’s comment: Results and Discussion, page 5 line 191: I guess that the word “result” has to be changed to “results”.
    Author’s Response: Thank you reviewer for your pin pointing. We had changed the word

  3. Reviewer’s comment: Results and Discussion, page 5 lines 198-199: The author has to link this sentence with their results. Do they obtain the expected system? If yes, which is/are the selected system(s).
    Author’s Response: Thank you reviewer for your comment. We have refined to link the sentences with results.

  4. Reviewer’s comment: Results and Discussion, page 6 lines 214-215: To my opinion, the sentence “Biocompatibility … application [24].” needs to be reworded.
    Author’s Response: Thank you reviewer for your comment. We have reworded to clarify the intended meaning.

  5. Reviewer’s comment: Results and Discussion, page 6 line 217: To my opinion, “exhibited non-cytotoxicity” has to be changed to “exhibited no cytotoxicity” or “are non-cytotoxic”.
    Author’s Response: Thank you reviewer for your comment. This part was corrected as per your suggestion.

  6. Reviewer’s comment: Results and Discussion, page 6 lines 218-219: Where is described this crystal violet staining experiment? Why does the author conclude that the crystal violet staining result was consistent with the CCK-8 results? The author has to develop and to explain his results and why they are consistent.
    Author’s Response: Thank you reviewer for your comment. We modified the 'figure 4b' position to avoid confusion. Both experimental methods are methods of measuring cell viability. However, the principle of the methods are different. CCK-8 assay is a measurement method based on mitochondrial activity. Crystal violet staining method directly stains cells and measures them based on the number of cells. In this study, first CCk-8 assay method was used to measure viability. In addition, crystal violet assay was additionally performed to confirm the CCK-8 result. The pattern of two similar outcomes convinces their viability outcome.

  7. Reviewer’s comment: Results and Discussion, page 7 lines 241-243: The sentence “Furthermore … ablation (Fig 6a).” needs to be slightly modified.
    Author’s Response: Thank you reviewer for your comment. We have modified as per your suggestion.

  8. Reviewer’s comment: Conclusions, page 7: I agree that the encapsulation of bioactive molecules into NPs can be important and interesting. However, why does the author want to encapsulate a bioactive molecule and to which kind of bioactive molecule the author think?

    Author’s Response: Thank you reviewer for your comment. We can encapsulate drugs to expect the synergistic effect of nanoparticles. For example, anticancer drugs such as doxorubicin can be loaded, and the anticancer drugs can be released through NIR stimulation. And they will generate heat from nanoparticles and NIR. Therefore, we described the drug encapsulation because we can expect more synergistic anticancer effects from the combination of these simultaneous effects.

Reviewer 3 Report

It is exciting to see that the authors are exploring on Polythiophene nanoparticles for anti-cancer therapy through NIR-triggered hyperthermal effect. The manuscript has presented the possible murine colorectal cancer killing effect when exposed to polythiopene nanoparticles and treated with NIR 808 nm. There are some major questions that have to be addressed to support the demonstration in the manuscript before acceptance. 

1. In section 2.2, the preparation method for polythiophene nanoparticles needs to be clarified. For example, "decimolar acetonitrile solution" of which compound? Thiophene or thiophene with ammonium persulphate? Better to use numerical numbers rather than "semi" or "deci". How the washing with methanol in acetonitrile solution was performed? What's the brand of centrifuge? Better to use RCF instead of RPM. What happened to the supernatant? Discarded or dried at room temperature? 

2. In terms of the cancer model, why author was selecting murine colorectal cancer cells CT26 rather than human colorectal cancer cells HT29, which has been widely used as a colorectal cancer model. 

3. In section 3.1, Figure 1b is too small to be visible. It's impractical to get the conclusion of particle sizes around 50-100 nm based on the TEM image. All particles are aggregated together. The author needs to provide DLS analysis to ensure no aggregation within nanoparticles obtained. 

4. In table 1, two concentrations were listed, 100 and 200 ug/ml, but the author didn't discuss why 100 ug/ml was picked over 200 ug/ml. The hyperthermal effect and cytotoxicity looks the same between them. Also, why not going lower than 100 ug/ml? 

5. In Figure 6, several justification should be included. First of all, Figure 6 b is too small to read and the FACS machine model/laser power, filters were not mentioned. Then, it looks like the PI fluorescence increased when cells are treated with NIR 808 nm. It raised up the question on the proper compensation in flow cytometry procedure. Most importantly, the author should consider about the annexin V and PI in differentiating early and late apoptosis. Otherwise, there is no meaning in using both dyes together. 

6. The most critical experiment on evaluating the therapeutic efficacy and penetrating efficiency of NIR 808 to the cancer targets should be discussed. Without the in vivo data, it's hard to prove the hypothesis. Can the author predict, discuss or address the concerns on in vivo efficacy, because the bio-distribution is the most critical part to make this strategy work. On line 56, 57, the author introduced the promising effect on metastasis tumor and immuno-memory effect, but didn't discuss anything in the following context. Thus, it's better not to raise this in the introduction. 

Other minor modifications should also be addressed: 

1. On line 38, 39, the reference for combination of oxaliplatin and 5-FU is missing. 

2. In material section, a lot of reagents were not specified. For example, information on the CCK-8 assay, livid/dead staining and the antibiotics used in media was missing. There are other types of reagents with missing information. Please check and include all those information. 

3. In section 2.4, all units are not consistent. 

4. In section 2.7, what's the lens used in microscopy?

5. Statistical analysis information is missing. 

6. On line 239, how did the author demonstrate the significance? From Figure 5c, it did show some red spots, but low in number. 

Author Response

Reviewer #3

It is exciting to see that the authors are exploring on Polythiophene nanoparticles for anti-cancer therapy through NIR-triggered hyperthermal effect. The manuscript has presented the possible murine colorectal cancer killing effect when exposed to polythiopene nanoparticles and treated with NIR 808 nm. There are some major questions that have to be addressed to support the demonstration in the manuscript before acceptance. 

  1. Reviewer’s comment: In section 2.2, the preparation method for polythiophene nanoparticles needs to be clarified. For example, "decimolar acetonitrile solution" of which compound? Thiophene or thiophene with ammonium persulphate? Better to use numerical numbers rather than "semi" or "deci". How the washing with methanol in acetonitrile solution was performed? What's the brand of centrifuge? Better to use RCF instead of RPM. What happened to the supernatant? Discarded or dried at room temperature? 
    Author’s Response: Thank you reviewer for your comment. We have clarified the method for the preparation of polythiophene in this revised manuscript. We have used numerical number to mention the concentration. E.g. M/10 solution of acetonitrile in water. Thiophene was dispersed into this M/10 acetonitrile solution to prepare M/2 solution of thiophene. We have also replaced the word RPM (Rotation per minute) by RFC (relative centrifugal forces).

  2. Reviewer’s comment: In terms of the cancer model, why author was selecting murine colorectal cancer cells CT26 rather than human colorectal cancer cells HT29, which has been widely used as a colorectal cancer model. 
    Author’s Response: Thank you reviewer for your comment. The CT26 cell line is widely used as injected cancer cell for subcutaneous formation of tumor tissue in mouse model. Although in vivo mouse model experiments were not performed in this study, we selected CT26 cell as a cancer cell model for further in vivo mouse model experiment.

  3. Reviewer’s comment: In section 3.1, Figure 1b is too small to be visible. It's impractical to get the conclusion of particle sizes around 50-100 nm based on the TEM image. All particles are aggregated together. The author needs to provide DLS analysis to ensure no aggregation within nanoparticles obtained. 
    Author’s Response: Thank you reviewer for your comment. As per your suggestions, we did DLS analysis and found the average size of the particle as 69.8 nm. More detail is presented in result and discussion section.

  4. Reviewer’s comment: In table 1, two concentrations were listed, 100 and 200 ug/ml, but the author didn't discuss why 100 ug/ml was picked over 200 ug/ml. The hyperthermal effect and cytotoxicity looks the same between them. Also, why not going lower than 100 ug/ml? 
    Author’s Response: Thank you reviewer for your comment. Nanoparticle is expected to have a hyperthermal effect by only NIR, therefore it should not be toxic in its normal condition. However, at the concentration of 500 ug/mL, the toxicity of the cells was observed due to the high concentration, so the experiment was not considered for concentration more than 200 ug/mL. In addition, in order to give hyperthermia more high power NIR laser should be used at low concentration of 100 ug/mL or less concentration. However, too high power NIR laser cannot be used in clinical application because of possible damage caused by its own NIR high energy. Therefore, in the table, experiments were conducted based on the two concentrations with the most suitable potential.

  5. Reviewer’s comment: In Figure 6, several justification should be included. First of all, Figure 6 b is too small to read and the FACS machine model/laser power, filters were not mentioned. Then, it looks like the PI fluorescence increased when cells are treated with NIR 808 nm. It raised up the question on the proper compensation in flow cytometry procedure. Most importantly, the author should consider about the annexin V and PI in differentiating early and late apoptosis. Otherwise, there is no meaning in using both dyes together. 
    Author’s Response: Thank you reviewer for your comment. We have described the FACS device model and related filter information in this revised manuscript. We corrected the figure 6b. We additionally described the results of apoptotic cell type and necrotic cell type in results and discussion section.

  6. Reviewer’s comment: The most critical experiment on evaluating the therapeutic efficacy and penetrating efficiency of NIR 808 to the cancer targets should be discussed. Without the in vivo data, it's hard to prove the hypothesis. Can the author predict, discuss or address the concerns on in vivo efficacy, because the bio-distribution is the most critical part to make this strategy work. On line 56, 57, the author introduced the promising effect on metastasis tumor and immuno-memory effect, but didn't discuss anything in the following context. Thus, it's better not to raise this in the introduction. 

    Author’s Response: Thank you Reviewer for your comment and suggestions. We have deleted the line of 56-57 of previous form of article regarding immune-memory effect. Although in vivo experiments were not conducted in this study, it is generally known that nanoparticles pass through the loosed blood vessel walls of cancer tissues and accumulate in cancer tissue. Therefore, according to the theory of general nanoparticle-sized scale particle’s accumulation, this study can suggest the possibility of cancer treatment using PTh-NPs by NIR.

    Reviewer’s comment: Other minor modifications should also be addressed: 

  7. Reviewer’s comment: On line 38, 39, the reference for combination of oxaliplatin and 5-FU is missing. 
    Author’s Response: Thank you Reviewer for your suggestions. We have to add the reference.

  8. Reviewer’s comment: In material section, a lot of reagents were not specified. For example, information on the CCK-8 assay, livid/dead staining and the antibiotics used in media was missing. There are other types of reagents with missing information. Please check and include all those information. 
    Author’s Response: Thank you reviewer for your suggestions. We corrected the missing information of reagents.

  9. Reviewer’s comment: In section 2.4, all units are not consistent. 
    Author’s Response: We have corrected the units.

  10. Reviewer’s comment: In section 2.7, what's the lens used in microscopy?
    Author’s Response: Thank you reviewer for your suggestions. We added the information of microscopy which is (Eclipse Ts2-FL, Nikon, Tokyo, Japan).

  11. Reviewer’s comment: Statistical analysis information is missing. 
    Author’s Response: Thank you reviewer for your suggestions. We added the statistical analysis section.

  12. Reviewer’s comment: On line 239, how did the author demonstrate the significance? From Figure 5c, it did show some red spots, but low in number. 
    Author’s Response: In the analysis by Live/Dead staining, the green color represents living cells and the red color represents dead cells or unhealthy cells. In the PTh-NP/NIR (808 nm) group, the number stained green cells was remarkably decreased because completely dead cells were detached from the bottom of the cell culture dish and fell off during the washing process. In addition, stained with red cells is unhealthy or still attached dead cells. Therefore, when these two observations are considered, it can be seen that cell death was clearly increased in PTh-NP/NIR (808 nm) group compared to other experimental groups.

Round 2

Reviewer 1 Report

The authors improved the paper according to all my comments, therefore I suggest accepting the manuscript in the current form.

Reviewer 2 Report

The authors have answered correctly to all the questions raised by the reviewers and improved their manuscript as asked.

Therefore, I do recommend the publication of this manuscript in Cells in the present form.

Reviewer 3 Report

All concerns have been addressed. Thank you authors for responding point by point. The only suggestion I have is further rearranging the Figure 6, so that Figure 6 b can be properly visualized. Other than that, I think the manuscript can be accepted in the present form.